# IgE and IgG4 Epitopes of Dermatophagoides and Blomia Allergens before and after Sublingual Immunotherapy

**DOI:** 10.3390/ijms24044173

**Published:** 2023-02-20

**Authors:** Daniele Danella Figo, Priscilla Rios Cordeiro Macedo, Gabriele Gadermaier, Cesar Remuzgo, Fábio Fernandes Morato Castro, Jorge Kalil, Clovis Eduardo Santos Galvão, Keity Souza Santos

**Affiliations:** 1LIM-19, Hospital das Clinicas HCFMUSP, Faculdade de Medicina, Universidade de Sao Paulo, Sao Paulo 05403-900, Brazil; 2Laboratorio de Imunologia, INCOR, Hospital das Clinicas HCFMUSP, Faculdade de Medicina, Universidade de Sao Paulo, Sao Paulo 05403-900, Brazil; 3Departamento de Clinica Medica, Disciplina de Imunologia Clinica e Alergia, Faculdade de Medicina da Universidade de Sao Paulo, Sao Paulo 01246-903, Brazil; 4Servico de Imunologia Clinica e Alergia, Hospital das Clinicas HCFMUSP, Faculdade de Medicina, Universidade de Sao Paulo, Sao Paulo 05403-000, Brazil; 5Department of Biosciences and Medical Biology, Paris Lodron University Salzburg, 5020 Salzburg, Austria; 6Instituto Nacional de Ciencia e Tecnologia de Investigação em Imunologia (iii-INCT), Sao Paulo 05403-900, Brazil

**Keywords:** IgE, IgG, house dust mites, immunotherapy, epitopes

## Abstract

Sublingual immunotherapy (SLIT) is used worldwide to treat house dust mites (HDM) allergy. Epitope specific immunotherapy with peptide vaccines is used far less, but it is of great interest in the treatment of allergic reactions, as it precludes the drawbacks of allergen extracts. The ideal peptide candidates would bind to IgG, blocking IgE-binding. To better elucidate IgE and IgG4 epitope profiles during SLIT, sequences of main allergens, Der p 1, 2, 5, 7, 10, 23 and Blo t 5, 6, 12, 13, were included in a 15-mer peptide microarray and tested against pooled sera from 10 patients pre- and post-1-year SLIT. All allergens were recognized to some extent by at least one antibody isotype and peptide diversity was higher post-1-year SLIT for both antibodies. IgE recognition diversity varied among allergens and timepoints without a clear tendency. Der p 10, a minor allergen in temperate regions, was the molecule with more IgE-peptides and might be a major allergen in populations highly exposed to helminths and cockroaches, such as Brazil. SLIT-induced IgG4 epitopes were directed against several, but not all, IgE-binding regions. We selected a set of peptides that recognized only IgG4 or were able to induce increased ratios of IgG4:IgE after one year of treatment and might be potential targets for vaccines.

## 1. Introduction

Allergen specific immunotherapy (AIT) is the only treatment capable of modifying the course of allergic diseases and altering its natural history, by stimulating allergen-specific immune tolerance [1]. It aims to reduce the degree of sensitization, and consequently, the characteristic symptoms of allergy preventing the progression from mild forms of allergy (rhinoconjunctivitis) to more serious diseases, such as asthma [2,3]. The treatment is initially marked by an increase in IgE, followed by an increase in IgG4 and subsequent reduction of IgE, with a continuous increase in specific IgG4 [4]. To treat allergic rhinitis (AR), AIT is performed mainly by the subcutaneous or sublingual route, for both children and adults. The sublingual route has been shown to be safe and effective, with lasting effect after its interruption [5,6].

House dust mites (HDM) are among the most important allergen sources containing many different allergenic molecules that are identified as etiological agents of allergic rhinitis in the Brazilian population [7]. The most common sensitizing species are *Dermatophagoides pteronyssinus*, *Dermatophagoides farinae,* and *Blomia tropicalis* [8,9]. For *Dermatophagoides pteronyssinus,* Der p 1, Der p 2, Der p 5, Der p 7, Der p 23 and Der p 21 seem to represent the most clinically relevant allergens. On the other hand, although Der p 10, a tropomyosin, is considered a major allergen in sources such as shrimp and cockroaches [10,11], it is considered a minor allergen in populations lacking exposure to cockroaches; however, this is different in Brazil, but epidemiological data is very scarce. *Blomia tropicalis* is a mite that was first described as a storage mite and it is now considered a house dust mite in tropical and sub-tropical areas, such as Brazil. It is less extensively studied and Blo t 5 is its major allergen [12].

There is a plethora of literature that evaluates molecular profiles of IgE and IgG4 recognition during AIT [13,14], especially subcutaneous. Epitope mapping goes back approximately 30 years [15], some previous works have identified epitopes in HDM allergens, such as Der p 5 [16], Der p 7 [17], Der p 2 [18,19], Der p 39 [20], and Blo t 11 [21] to characterize IgE only or IgE and IgG binding sites by various methods. Other studies used epitope mapping with the objective of investigating antibody cross-reactivity among different molecules [22,23,24]; Yi et al. evaluated similarities between Blo t 10 and Der p 10 [25]. Already in 1996, Kobayashi et al. [26] compared epitopes recognized by Der p 2-specific IgE antibodies with those recognized by IgG4 antibodies, substantiating their differences, and pointing out that similar studies mapping epitopes after immunotherapy would be crucial for understanding these differences. Some studies have evaluated these differences for food allergens [27,28]. However, studies revealing changes of epitope binding during mite AIT are lacking.

Due to advances in molecular allergology, new allergy vaccines have been developed, among those are peptide vaccines. Immunotherapy with a recombinant B-cell epitope-based, hypoallergenic, vaccine for grass pollen allergy has demonstrated its safety and efficacy [29,30], and in a double-blind placebo-controlled trial, it has been proven to diminish allergic symptoms of patients [29]. Given the crucial prerequisite for IgE-blocking antibodies for long-term clinical efficacy, future studies are needed to determine whether this requires continuing affinity maturation of B cell IgG responses on continued exposure or increased avidity of blocking antibodies due to augmented polyclonality [31]. Some possible benefits of peptide vaccines are that they may not induce IgE- or T cell-mediated side effects, they have high specificity, low ability to promote allergen-specific IgE responses, and the technology can be applicable for many allergens [32].

Until recently, dynamic epitope profiles during immunotherapy were not well defined. In this study, we present a first approximation analysis of IgE and IgG4 epitope binding to ten HDM allergens: Der p 1 (cysteine protease), Der p 2 (NPC2 family), Der p 5, Der p 7 (Bactericidal permeability-increasing-like protein), Der p 10 (Tropomyosin), Der p 23 (Peritrophin-like protein domain), Blo t 5, Blo t 6 (Chymotrypsin), Blo t 12, and Blo t 13 (Fatty acid-binding protein) using sera from ten patients undergoing sublingual immunotherapy (SLIT) for allergic rhinitis. Immunotherapy treatments are immune modulatory, and increasing tolerance is associated with increases in IgG4:IgE ratios [33]. Therefore, we aim to better elucidate the IgE and IgG4 epitope profiles during SLIT and identify short peptides able to induce increasing ratios of IgG4:IgE in allergic individuals.

## 2. Results

To analyze IgE and IgG4 peptide recognition we selected ten relevant HDM allergens (i.e., Der p 1, Der p 2, Der p 5, Der p 7, Der p 10, Der p 23, Blo t 5, Blo t 6, Blo t 12, and Blo t 13) for epitope mapping. A serum pool of 10 patients undergoing SLIT for house dust mite allergy was used for testing before and 1 year after treatment. All patients reported a self-perception of improvement of rhinoconjunctivitis symptoms. Demographical data of individuals included together with clinical symptoms and IgE titers are provided in Table 1.

Peptides with Median Fluorescence Intensity (MFI) higher than 100 were considered positive. Diversity in peptide reactivity was higher post-1-year for both antibodies, going from 46 IgE-reactive to 60 and from 19 to 66 IgG4-reactive peptides (Figure 1a).

Peptide diversity for IgE varied among molecules and timepoints without a clear pattern, similar to what was observed for total and specific IgE levels (Table 1).

Considering allergens individually, Der p 10 showed the highest IgE peptide diversity pre-SLIT, followed by Der p 1 (Figure 1b). For *Blomia tropicalis*, the molecule with the highest IgE peptide diversity detected at pre-treatment was Blo t 12 (Figure 1d). All allergens, except Der p 5 and Der p 23, presented IgG4 peptide binding before SLIT. Again, Der p 10 was the molecule presenting more peptides recognized, and among Blo t allergens, all presented reacting peptides (Figure 1c,e).

During AIT, we considered there to be an increase or decrease in peptide reactivity if there was a difference of at least 20%. Post-1-year, Der p 1, Der p 5, and Der p 23 (Figure 1b) had more IgE-peptides after treatment than before. On the other hand, Der p 2, Der p 7, and Der p 10 (Figure 1b) had lower diversity post-1-year. Der p 5, similar to Blo t 13, presented no IgE binding peptides at basal timepoint but de novo sensitization to two peptides of these allergens after 1-year of SLIT was shown in our serum pool (Figure 1b,d).

Peptide diversity recognized by IgG4 were higher for all Der p tested molecules post-1-year, except for Der p 1, which showed one less peptide after SLIT (Figure 1c).

Blo t 5 and Blo t 6 both kept the same number of peptides before and after treatment (Figure 1d). A new sensitization to Blo t 13 was shown post-1-year (Figure 1d). IgG4 peptide diversity was higher for all tested molecules post-1-year (Figure 1e).

Regarding IgG4 profiles, patients undergoing treatment clearly demonstrated a higher and broadening diversity in reactivity to all allergens except Der p 1. This observation thus seems to be independent of pre-existing IgG4 reactivity, which was noted for most allergens (Figure 1c,e).

Considering the median fluorescence intensities (MFI) to individual allergen peptides, IgE reactivity to *Dermatophagoides pteronyssinus* allergens mostly increased, while it mostly decreased to *Blomia tropicalis* allergens after treatment. However, this observation was statistically significant only for Der p 1 (Figure 2a). As for IgG4, an increase in amplitude of response after treatment was observed for the molecules Der p 5, Der p 7, Der p 10, Blo t 12, and Blo t 13 (Figure 2b). In line with the enhancement of IgG4 epitope diversity, SLIT also induced a higher antibody binding capacity to the allergen peptides.

To better understand the dynamic profile of peptide recognition by IgE and IgG4, patterns of bindings before and after 1-year SLIT were compared (Figure 3). Some peptides were IgE-reactive at basal timepoint and were not detectable after 1-year SLIT, such occurred for Der p 1, 2, 7, 10 and 23, and also for Blo t 6. Other peptides were recognized similarly or with higher intensity, e.g., Der p 2 and Der p 23. Several additional IgE-reactive peptides were detected for Der p 1, 2 and 23, and Blot t 5, 6, and 12, during treatment. Blo t 13 did not present any IgE-reactive peptides before therapy but after 1-year SLIT, two novel peptides were detected. Most molecules presented higher diversity of IgG4-reactive peptides after one year. Interestingly, Der p 5, which did not display any detectable peptides pre-treatment, presented de novo IgE and IgG4-reactive peptides in the next timepoint. The majority of identified IgG4 peptide regions co-localized at least partially with IgE-binding peptides. However, some IgG4 peptides derived from Der p 5, Blo t 5, and Blo t 13, for example, seemed to be unique, mostly lacking overlap with IgE peptides.

To better understand whether the IgG4 that was produced during immunotherapy has similar specificity to the original sensitization profile, post-immunotherapy IgG4 binding patterns were compared to baseline IgE binding patterns. We therefore used three-dimensional models of the individual allergens (Figure 4). Identified epitopes were typically located at exposed regions of the proteins. Several unique IgE epitopes shown in yellow were detectable on all allergens except Der p 5 and Blo t 13. Unique IgG4 epitopes, in blue, were detected on all molecules except Der p 1, with most in concordance with IgE, showing that IgG4 can block IgE activity by competing for the binding region. All allergens, excluding Der p 5 and Blo t 13, presented overlapping pre-existing IgE and post-SLIT IgG4 regions, highlighted in pink. Based on our peptide data, SLIT-induced IgG4 antibodies recognized either unique or overlapping IgE binding regions for all allergens.

Besides the dynamics of peptide recognition during SLIT, we aimed to find potential targets for peptide-based vaccines. We focused on regions that exclusively elicited IgG4 during treatment without co-localization to pre-existing or post-IgE reactivity and also on peptides that induced higher ratios of IgG4:IgE after one year of therapy compared to pre-treatment (Table 2). For unique IgG4 regions we considered all peptides with at least 12 residues. One to three peptides with a length up to 33 residues could be identified for Der p 5 and 7 and to all Blo t allergens, that exclusively elicited an IgG4 response. We found 25 peptides with this ability that corresponded to 19 consent epitopes. All molecules tested presented at least one epitope selected (Table 2).

## 3. Discussion

For allergic rhinitis, subcutaneous immunotherapy (SCIT) has been the gold standard. However, in recent years, sublingual immunotherapy (SLIT) has emerged as an effective and safer alternative. There are many studies on immunological profiles developed during SCIT, but for SLIT there is still little information available [34].

Although several studies have mapped IgE and/or IgG epitopes for HDM allergens, [16,17,18,19,21,22,23,24,25], studies that evaluate dynamics of epitope binding over the course of allergen immunotherapy are missing. Differences in IgE or IgG4 antibody specificities have also been reported for other allergens, specifically Der p 1 [35] and Der p 2 [26].

To our knowledge this is the first study to evaluate epitope binding for HDM allergens before and after one year of immunotherapy. To do so, we used a peptide microarray and investigated the diversity and amplitude of IgE and IgG4 peptide repertories against ten HDM allergens. Pooled sera from ten patients undergoing allergen immunotherapy were used at a basal timepoint and after 1-year of SLIT. From the 544 peptides analyzed in the microarray, 46 and 60 IgE peptides as well as 19 and 66 IgG4 peptides were identified before and after 1-year SLIT, respectively. Overall, half of the allergens presented a decreased or similar number of IgE peptides after one year, while all allergens, except Der p 1, presented a higher number of peptides being recognized by IgG4 after 1-year SLIT. Same as total and specific IgE levels, degrees of recognition varied among molecules, antibodies, and timepoints without a clear tendency, probably due to the short interval of immunotherapy.

While for most Der p allergens IgE reactivity after one year was found to be similar to baseline levels, several relevant allergens including Der p 1, Der p 2, and Der p 23 showed enhanced reactivity to several peptides. Given the short time span of SLIT, considering the full treatment lasts three years, an increase in IgE reactivity can be anticipated at least for some allergens. A previous study comparing SCIT and SLIT showed that clinical and immunological improvements started after one year of SCIT but only after two years of SLIT [34]. Similarly, an increase in IgE epitope diversity during four years of oral immunotherapy against peanuts was observed by Vickery et al. [28]. In addition, specific IgE levels against Der f 1, Der p 10, 11, and 23 were reported to remain unchanged over the course of 12 months of immunotherapy [36].

One study analyzed IgE epitopes in *Dermatophagoides farinae* allergens, using methodology similar to our work, identifying four peptides for Der f 1, three for Der f 2, two for Der f 5, and three for Der f 7. Similar to our study, all these epitopes were in exposed areas on the surface of the molecules, which would allow the epitopes to recognize the IgE antibody [37].

Besides IgE, we were particularly interested in the identification of IgG4 epitopes induced during AIT. The observed increase in IgG4 binding to HDM peptides/epitopes as well as in IgG4 amplitude during SLIT in our study is in line with previous observations of increased allergen-specific IgG4 levels during immunotherapy for house dust mite as well as other allergen sources [13,14,38].

IgG4 antibodies are considered to compete with specific IgE for binding to the allergen. This might inhibit IgE-mediated complex formation on sIgE receptor-expressing cells, such as mast cells and basophils [39].

Immunotherapy was shown to alter epitope binding patterns, causing a polyclonal increase in IgG4 levels, with a simultaneous reduction in intensity of IgE epitopes [28]. Accordingly, in our study, higher IgG4 peptide diversity was observed after one year of SLIT for all allergens, including only minor changes for Der p 1 but also de novo induction of IgG4 reactivity. IgE intensities varied as mentioned above.

Interestingly, Der p 10, a tropomyosin, was the allergen with most IgE and IgG4 peptides being recognized. This protein is typically described as a minor allergen in the temperate climate areas but is considered a major allergen in other sources such as shrimps, helminths, and cockroaches [10,11,40]. Epidemiological data on Der p 10 sensitization are very scarce in Brazil. Given the observed high reactivity to Der p 10 peptides, further studies are needed to clarify its role in HDM allergy, since there is a concomitant exposition to tropomyosin-containing cockroaches and helminths in our country [41,42]. In a study conducted in Zimbabwe, that is also located in a tropical region with a population highly exposed to cockroaches and crustacea, 55% of patients presented specific IgE to Der p 10 [43], showing the importance of conducting allergen studies in different populations. In addition, Blo t 12 represented the allergen with the most linear IgE epitopes from this mite before and after treatment, not Blo t 5 as was expected since it is considered the most clinically relevant allergen of *Blomia tropicalis* [44]; thus, further investigations of Blo t 12 on the Brazilian population are also warranted.

Some major HDM allergens, such as Der p 2 and Der p 23, presented only few linear IgE epitopes. Der p 2, one of the most clinically important allergens, was one of the molecules with few IgE and IgG4 epitopes being detected. This may be explained by the fact that many Der p 2 epitopes are mostly defined by structural features formed from discontinuous regions of the allergen, rather than by a continuous peptide sequence. IgE binding is known to be vulnerable to reduction and alkylation, showing the presence of many conformational epitopes, especially for Der p 2 [45].

It has been postulated that the use of whole extract for allergen immunotherapy could induce neo sensitizations during treatment [46], similar to our observation for Der p 5 and Blo t 13. Notably, this was also observed for pollen allergen immunotherapy after one year [47].

As in our study we were highly interested in the therapy-induced IgG4 antibodies, we have chosen the 1-year timepoint to identify the first IgG4 epitopes in investigated HDM allergens. We aim to follow this evaluation in later timepoints of treatment to better understand the dynamics of epitope recognition. Indeed, even after this first period, the IgG4 diversity and reactivity level was already higher in most allergens, in line with previous observations [48]. The majority of herein identified IgG4 epitopes at least partially overlapped with linear IgE epitopes, thus they are most likely blocking IgE by allosteric inhibition and competing for the same allergen binding site. While this represents one of the hallmarks of successful AIT, this also indicates a success in therapy when after one year of treatment there was a self-perception of a better clinical control of the rhinoconjunctivitis symptoms.

To investigate regions that could potentially be used for peptide immunotherapy, we used two approaches to obtain a list of candidate targets to be further evaluated. First, we focused on IgG4 peptides that did not co-localize with IgE epitopes. An engineered vaccine made of non-IgE binding peptides fusioned to the hepatitis B preS protein was able to induce allergen-specific IgG, improve clinical symptoms of seasonal grass pollen allergy, and was well tolerated [29].

Regions that exclusively elicited IgG4 during treatment showed epitopes from 12 up to 33 residues. That length is in the middle of existing data for peptide vaccines, where IgE-mediated peptides are longer, being 20–40 amino acids and typically dependent on a folded tertiary structure [49,50]. Shorter peptides, 10–17 aa, are mainly T-cell epitopes which are designed to be recognized by MHC-class II molecules. Furthermore, those shorter peptides should be unable to bind IgE-FcεRI on effector cells due to their small size [50].

One of the identified IgG4 exclusive epitopes, Der p 5_62–94_, comprehends a region which was previously predicted as a conformational B cell epitope and potential target for a vaccine design for Der p 5: ^77^EKPTKEMKD^85^ [51]. Although most B cell epitopes are longer and conformational, ten percent are linear, and those linear epitopes may also be involved in conformational epitopes [52].

Next, based on the known ability of allergen immunotherapy to induce IgE-blocking antibodies, we aimed to further evaluate regions recognized by both antibodies at each timepoint. IgG4 monoclonal antibodies directed against IgE epitopes of major allergens have proved successful in inhibiting human nasal allergen challenge responses after injection of cocktails of passive immunotherapy [53]. Since increased ratios of Ig4:IgE are observed after tolerance, we selected peptides that presented higher ratios of IgG4:IgE after one year of therapy compared to pre-treatment, and that comprised 19 epitopes from 15 up to 62 residues. Higher ratios of sIgG4 to sIgE were significantly associated with immunotherapy success in a group of peanut allergic patients [33]. We therefore raise the hypothesis that peptides able to induce higher IgG4:IgE ratios could be potential targets for future studies on immunotherapy. From the set of peptides that we identified, some of them have already been reported as potential B cell epitopes using an in silico approach, reinforcing our proposal [51].

All peptides identified in our study were tested against predicted T and B cell epitopes using the Immune Epitope Database (https://www.iedb.org/, accessed on 2 February 2023) [54] and none were a match.

We are now beginning to understand the mechanisms of standard and novel approaches to immunotherapy. We suggest that peptide immunotherapy will significantly improve the safety and management of allergies. The challenges arising from current approaches, including the risk of side-effects, onerous duration of treatment, poor adherence, and high cost can be overcome by application of peptides based on T or B cell epitopes, rather than whole allergens [55]. While whole allergen or recombinant wild type-like allergens induce IgE responses and may stimulate IgE sensitization, modified allergens or hypoallergenic peptides are unlikely to induce IgE and could play an important role in the future for primary prevention [56].

In summary, results herein presented revealed that IgE profile is variable, with increased levels detected for several peptides of some molecules after one year of treatment. IgG4 recognition was always increased showing that mite allergic patients receiving SLIT can elicit IgG4 to several epitopes from ten mite allergens partially overlapping with IgE after one year of treatment. Some unexpected results revealed that Der p 10 and Blo t 12 were the molecules with the highest number of peptides being recognized, indicating those allergens may play an important role in the Brazilian population. We show that the repertoire for linear IgE and IgG4 epitopes is not identical, and some peptides could be selected as potential vaccine candidates aiming to circumvent IgE-related side effects. For this, we have chosen peptides that are IgG4 exclusive or that induced higher IgG4:IgE ratios, resulting in a set of 29 epitopes. Although this study consists of only a small number of patients, this is the first report considering a panel of different allergens from two different mites in two timepoints of SLIT. Considering the paucity of information on epitope profile during SLIT, the results shown here can pave the way for future studies on specific peptides that can be useful as vaccine candidates.

## 4. Materials and Methods

### 4.1. Patient Selection

Based on the clinical history and samples available, 10 patients with persistent allergic rhinitis (>4 days per week and >4 weeks) and moderate–severe symptoms (impairment of daily activities) who were undergoing a SLIT (sublingual immunotherapy) protocol study with IPI-ASAC (Alicante, Spain) extract for Der p and Blo t were selected at the Ambulatório de Imunologia Clínica e Alergia do HCFMUSP (Hospital das Clínicas da Faculdade de Medicina da Universidade de São Paulo) (Table 2). Inclusion criteria consisted of sensitization to *Dermatophagoides pteronyssinus* and *Blomia tropicalis* (confirmed by prick test with papule ≥3 mm or specific serum IgE and normal spirometry (FEV1 ≥70% of predicted)).

The treatment of allergic rhinitis was standardized for all participants, according to their clinical condition. The following medications were used: budesonide nasal spray as control medication and loratadine 10 mg/day per ors as rescue medication. As for the patients who presented with associated allergic conjunctivitis, the use of olopatadine eye drops was indicated.

Pooled sera before treatment (pre-SLIT) and after one year of immunotherapy (post-1-year) were used in this study. This research was approved by the “Comissão de Ética para Análise de Projetos de Pesquisa do Hospital das Clínicas da Faculdade de Medicina da Universidade de São Paulo” (CAAE: 64005516.1.0000.0068).

### 4.2. Epitope Microarray

Peptide microarray was produced by laser-printing (PEPperPRINT- Heidelberg, Germany). The amino acid sequences of 10 allergens were identified: Der p 1 (P08176.2), Der p 2 (CAK22338.1), Der p 5 (P14004.2), Der p 7 (P49273.1), Der p 10 (O18416.1), Der p 23 (L7N6F8.1), Blo t 5 (O96870.1), Blo t 6 (AAQ24544.1), Blo t 12 (Q17282.1), and Blo t 13 (Q17284.1). GSGSGSG linkers at the C- and N-terminus were used to avoid truncated peptides. The linked and elongated antigen sequences were translated into 15 amino acid peptides with a peptide-peptide overlap of 12 amino acids. The resulting linear peptide microarrays contained 544 different peptides (Appendix A) printed in duplicate and were framed by additional HA (YPYDVPDYAG) and polio (KEVPALTAVETGA) control peptides. The array was incubated for 15 min in phosphate buffer saline with 0.05% Tween 20 (PBS-T), followed by an initial blocking for 30 min with Rockland Blocking Buffer MB-070 (RBB; Rockland Immunochemicals, Royersford, PA, USA). After short rinsing with PBS-T, the array was incubated for 16 h at 4 °C with human sera samples at 1:100 and 1:5 dilutions to IgG4 and IgE detection, respectively. The array was washed three times for 1 min with PBS-T and then incubated for 45 min at room temperature with secondary antibody Mouse anti-human IgG4 DyLight680 (Rockland Immunochemicals, Royersford, PA, USA) (0.1 µg/mL) or goat anti-human IgE (ε chain) DyLight800 (Rockland Immunochemicals, Royersford, PA, USA) (1 µg/mL). Subsequently, the array was washed three times for 1 min with PBS-T and rinsed with 1 mM TRIS, pH 7.4. Fluorescence images were acquired with an Odyssey Infrared Imager (LI-COR Biosciences, Lincoln, NE, USA) at a resolution of 21 µm, scanning offset 0.65 mm, and scanning intensities of 7/7 (red = 700 nm/green = 800 nm). Quantification of spot intensities and peptide annotation were based on the 16-bit grayscale tiff files that exhibit a higher dynamic range than the 24-bit colorized tiff files. Microarray image analysis was done with PepSlide^®^ Analyzer 2.0. A software algorithm breaks down fluorescence intensities of each spot into raw, foreground, and background signal, and calculates average median foreground intensities and spot-to-spot deviations of spot duplicates. A maximum spot-to-spot deviation of 40% was tolerated, otherwise the corresponding intensity value was zeroed. To increase specificity, values above 100 were considered positive for categorical analysis.

### 4.3. Statistical Analysis

The comparison of peptide intensity was conducted using the non-parametric paired samples Wilcoxon test. All statistical analyses were two-tailed, and the significance was set at *p* < 0.05. Analyses were performed using GraphPad Prism 9.4.1(GraphPad Software, San Diego, CA, USA). The cutoff value for IgE and IgG4 was established as the average of all measurements above zero at the initial time. To avoid disturbances from outliers, this average was a trimmed mean including the central 95% of the sample with value 100.35, considered 100.

### 4.4. Protein Modelling

Tridimensional structures of the allergens Der p 1 (5VPG), Der p 2 (1A9V), Der p 5 (3MQ1), Der p 7 (3H4Z), and Der p 23 (4ZCE) from *Dermatophagoides pteronyssinus* and Blo t 5 (2JMH) from *Blomia tropicalis* were taken from the RCSB Protein Data Bank.

The 3D structure of Blo t 12 (2MFK) of the RCSB Protein Data Bank only showed the chitin-binding domain corresponding to residues 59–124. To complete the structure of Blo t 12, the N-terminal region (1–58) was modelled using the trRosetta de novo structure prediction server (https://yanglab.nankai.edu.cn/trRosetta/, accessed on 14 October 2022 ) [57].

The structures of the remaining allergens, Der p 10, Blo t 6, and Blo t 13 were obtained by homology modeling using the MODELLER 10.3 software (California, San Francisco) and 7KO4, 2F91, 2A0A as templates, respectively. Saves 6.0 (https://saves.mbi.ucla.edu, accessed on 5th December 2022) was used to validate our models. All five models were run through Procheck to check the stereochemical quality of the structure by analyzing residue-by-residue geometry and overall structure geometry. The best model was chosen by Ramachandran plot results. Superimposition of query and template structure and visualization of generated models were performed using UCSF Chimera (California, San Francisco) software package 1.16 [58].

## Figures and Tables

**Figure 1 ijms-24-04173-f001:**
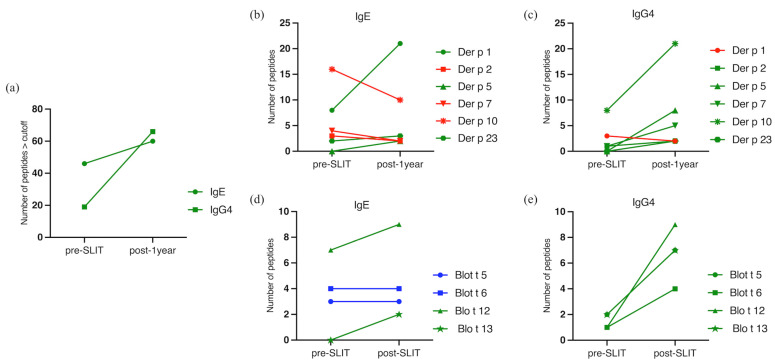
Diversity of peptide reactivity pre- and post-1-year SLIT. (**a**) Diversity of peptides pre- and post-SLIT for IgE and IgG4; (**b**) IgE peptide diversity for *Dermatophagoides pteronyssinus* molecules; (**c**) IgG4 peptide diversity for *Dermatophagoides pteronyssinus* molecules; (**d**) IgE peptide diversity for *Blomia tropicalis* molecules; (**e**) IgG4 peptide diversity for *Blomia tropicalis* molecules. Green lines indicate an increase of more than 20%, red lines indicate a decrease of at least 20%, blue lines were considered not altered (<20% of alteration).

**Figure 2 ijms-24-04173-f002:**
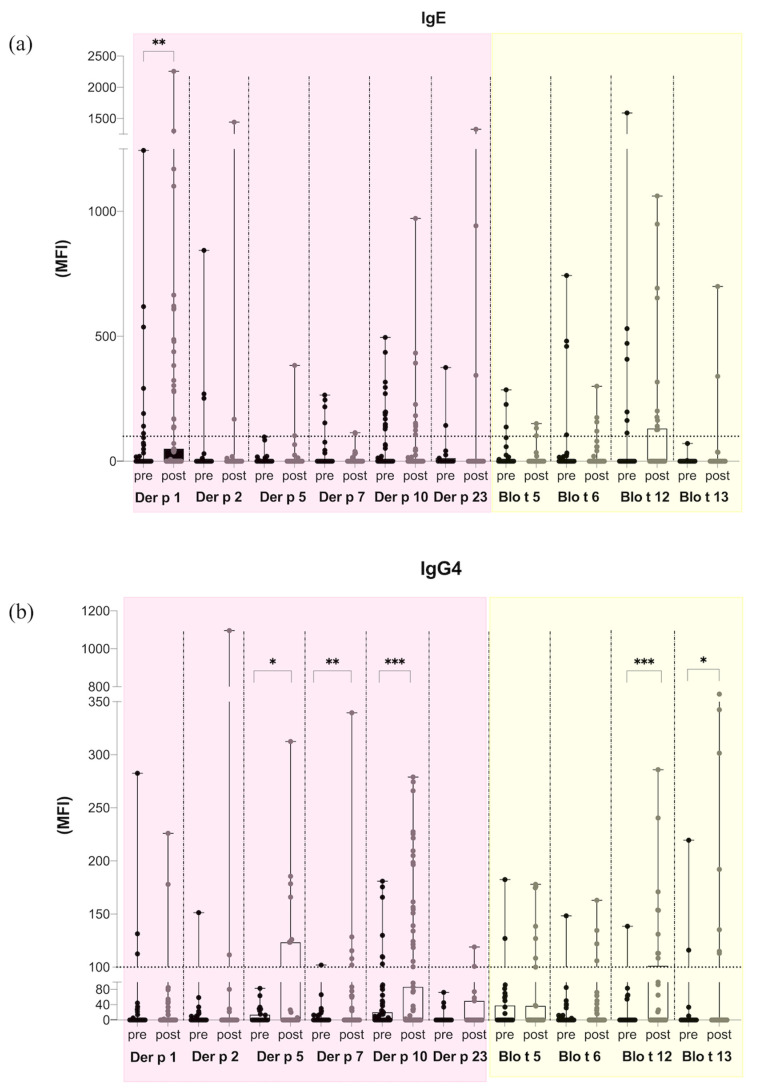
Intensity of antibody reactivity to *Dermatophagoides pteronyssinus* and *Blomia tropicalis* peptides analyzed for each allergen. (**a**) IgE amplitude pre- and post-1-year SLIT. (**b**) IgG4 amplitude pre- and post-1-year SLIT. Box plot, minimum to maximum showing all points. Wilcoxon test was made to compare pre and post intensities for each allergen. MFI = Median Fluorescence Intensity. * *p* < 0.05; ** *p* < 0.001; *** *p* < 0.0001.

**Figure 3 ijms-24-04173-f003:**
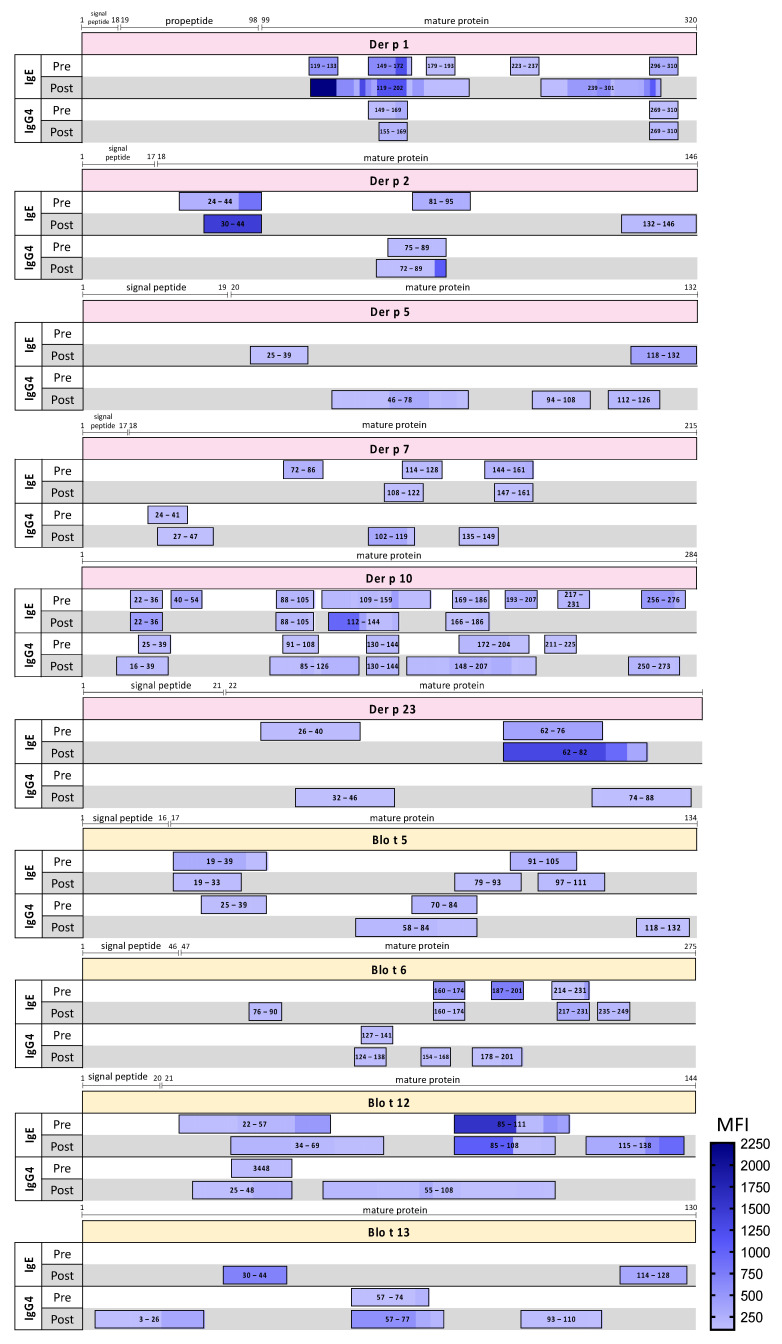
Epitopes’ localization and amplitude of responses for IgE and IgG4 pre- and post-1-year SLIT. Peptides’ positions were assigned considering the whole sequence of protein. MFI = Median Fluorescence Intensity.

**Figure 4 ijms-24-04173-f004:**
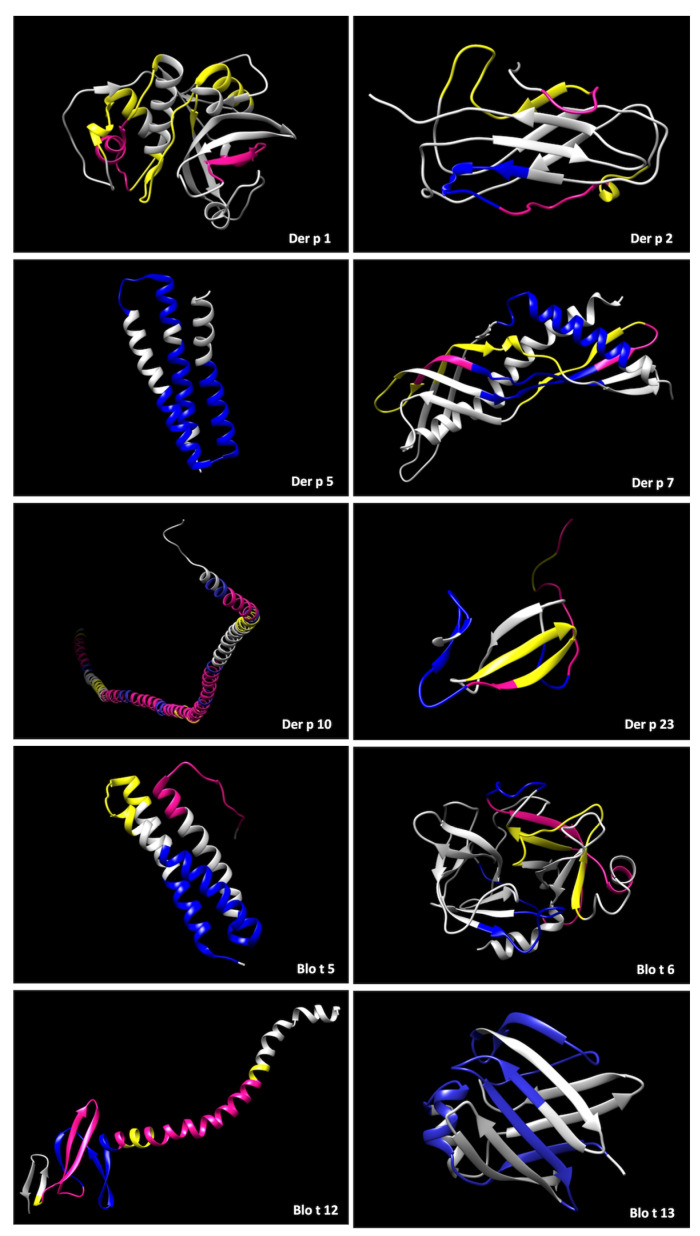
Protein modeling of tertiary structure of allergens highlighting peptides/epitopes detected in the microarray. In yellow, IgE pre-SLIT binding epitopes, in blue, IgG4 post-1-year binding epitopes, and in pink, regions where both antibodies have affinity.

**Table 1 ijms-24-04173-t001:** Demographic and clinical data from selected patients.

Patient	Sex	Age (y)	Symptoms	Pre-SLIT					Post 1-Year		
				IgE	IgE	IgE	Prick	Prick	IgE	IgE	IgE
				Total	Der p 1	Blomia	DPT	Blomia	total	Der p 1	Blomia
				(UI/mL)	(kUA/L)	(kUA/L)	(mm)	(mm)	(UI/mL)	(kUA/L)	(kUA/L)
4	F	48	AR, conjunctivitis	49.9	1.61	0.27	5	6	20.8	0.2	1.23
10	F	23	AR, conjunctivitis, asthma	2500	>100	>100	4	5	905	93.5	ND
16	F	23	AR, conjunctivitis	496	11.9	67	6	5	68.3	52.2	5.89
40	F	25	AR, asthma	19.3	0.26	11.2	5	6	36.2	0.06	8.11
44	F	43	AR, conjunctivitis	52.9	20.9	0.02	5	4	58.5	0.02	14
48	M	15	AR, conjunctivitis, dermatitis	324	29.9	16.9	9	5	194	13.9	23.6
51	M	15	AR, conjunctivitis, asthma, dermatitis	86.2	16.5	0.04	5	3	56.4	0.03	11.3
58	F	20	AR	28.3	90	>100	4	6	35	0.04	5.88
61	M	13	AR, conjunctivitis	59.4	1.43	19.6	5	4	98.8	20.6	0.89
77	M	15	AR, conjunctivitis	529	1.52	1.45	7	4	>100	9.77	11.4

Abbreviations: y, year; AR, allergic rhinitis; ND, not done.

**Table 2 ijms-24-04173-t002:** Potential targets for vaccine development. Selected peptides that are IgG4 unique or elicited increased IgG4:IgE ratios after one year of treatment.

Unique IgG4 Epitopes							
Allergen		Consent Epitope				Corresponding Aminoacid Position	
Der p 5		ELALFYLQEQINHFEEKPTKEMKDKIVAEMDTI				62–94	
		QRKDLDIFEQYNLEM				110–124	
Der p 7		EEINKAVDEAVAAIEKSETFD				27–47	
Blo t 5		DELNENKSKELQEKIIRELDV				58–78	
		LKDLKETEQKVKDIQ				118–132	
Blo t 6		VQHEQYDPNTIENDI				124–138	
Blo t 12		KTTTEETHHSDDLIV				70–84	
Blo t 13		IEGKYKLEKSDNFDKFLDELGVGF				3–26	
		KNTEIKFKLGEEFEEDRADGK				57–77	
		QTQYGDKEVKIVRDFQGD				93–110	
**Epitopes with increased igg4:ige ratios after treatment**							
**Allergen**	**Corresponding aminoacid position**	**Peptide**	**Ratio IgG4:IgE**		**Consent Epitope**		**Corresponding aminoacid position**
			**pre-SLIT**	**post-1 year**			
Der p 1	149–163	RNQSLDLAEQELVDC	0.21	83.5	RNQSLDLAEQELVDC		149–163
	296–310	DNGYGYFAANIDLMM	0.39	3.98	DNGYGYFAANIDLMMIEE		296–313
	299–313	YGYFAANIDLMMIEE	0.01	46.5			
Der p 2	24–38	DCANHEIKKVLVPGC	0.23	80	DCANHEIKKVLVPGC		24–38
Der p 5	38–53	KKHDYQNEFDFLLME	0.05	26.5	KKHDYQNEFDFLLME		38–53
Der p 7	27–41	EEINKAVDEAVAAIE	3.09	102	EEINKAVDEAVAAIE		27–41
	103–117	LLVGVHDDVVSMEYD	0.01	339.5	LLVGVHDDVVSMEYD		103–117
Der p 10	110–124	TAKLEEASQSADESE	0.12	221.5	TAKLEEASQSADESE		110–124
	146–160	NQLKEARMMAEDADR	0.01	76	NQLKEARMMAEDADRKYDEVARKLAMVEADLERAEERAETGESKIVELEEELRVVGNNLKSL		146–208
	149–163	KEARMMAEDADRKYD	0.17	161.5			
	170–184	AMVEADLERAEERAE	0.22	1.8			
	182–196	RAETGESKIVELEEE	13	34.88			
	194–208	EEELRVVGNNLKSLE	0.27	2.37			
	212–226	EKAQQREEAHEQQIR	1.55	86.5	EKAQQREEAHEQQIR		212–226
	257–271	EDELVHEKEKYKSIS	0.16	134	EDELVHEKEKYKSIS		257–271
Der p 23	74–88	KDCPGNTRWNEDEET	1.76	119	KDCPGNTRWNEDEET		74–88
Blo t 5	19–33	EHKPKKDDFRNEFDH	0.29	1.32	EHKPKKDDFRNEFDHLLIEQA		19–39
	25–39	DDFRNEFDHLLIEQA	0.93	3.96			
	70–84	EKIIRELDVVCAMIE	3.15	127	EKIIRELDVVCAMIE		70–84
Blo t 6	127–141	EQYDPNTIENDISLL	5.39	31	VQHEQYDPNTIENDISLL		124–141
	187–201	NLQVGELKIVSQEEC	0.11	2.02	NLQVGELKIVSQEEC		187–201
Blo t 12	22–36	DEQTTRGRHTEPDDH	0.48	92.5	DEQTTRGRHTEPDDH		22–36
	97–111	EEGPIHIQEMCNKYI	0.15	26.5	EEGPIHIQEMCNKYI		97–111
Blo t 13	12–26	SDNFDKFLDELGVGF	11.17	342.25	SDNFDKFLDELGVGF		12–26
	96–110	YGDKEVKIVRDFQGD	0.01	3.19	YGDKEVKIVRDFQGD		96–110

## Data Availability

Not applicable.

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
