# Peer review of "IgE and IgG4 Epitopes of Dermatophagoides and Blomia Allergens before and after Sublingual Immunotherapy"

_ijms, 2023, doi:10.3390/ijms24044173_

Round 1
Reviewer 1 Report
An interesting work on this new line of research on sensitization to mites with possible significance towards therapeutic uses
Limitations: the panel of mite allergens studied are limited and would be applicable only to some populations, especially for Blomia tropicalis.
The results should be presented only as a first approximation as they are carried out only with a limited sample of only 10 patients.
Reviewer 2 Report
The paper examines the IgE and IgG4 epitopes of Dermatophagoides and Blomia allergens before and after sublingual immunotherapy .
The topic is interesting but not completly new and was investigated for food allergens (peanut and milk) already.
Please include the references and change the sentence in lane 66 accordingly.
lane 116: What is the reason for this 20% limit?
Some questions: Why were the peptides spotted in duplicates and not in triplicates as known from the ISAC chip?
Why were the sera pooled and not tested as single sera, when the usage of serum is so low due to the high dilutions used.
Why did you not measure also IgG1?
Is there something known about the extract used for SLIT, especially if all the relevant allergens are really included in the preparation.
Minor points:
Table1: the first Derp 5 epitope is not very clear because of the line break
Please change Ref. 40
Reviewer 3 Report
The article ‘ IgE and IgG4 epitopes of Dermatophagoides and Blomia aller- 2
gens before and after sublingual immunotherapy’ submitted to the journal of IJMS The article is structured well, and the work is relevant for the readership of IJMS The authors have provided valuable information about the Sublingual immunotherapy advantages for their target hypersensitivity disorders.
However, some points “minor” can be improved or modified by the authors. It would help for the readers to have a final summary that delineates all the interactions discussed in this article. Also, to emphasize the new strategies for the treatment of these disorders.
(line 282): during the treatment period, have the participants received any medications, or vaccines, or performed any surgical operations? Please add any information this to your article.
Have you done any phenotypical analysis of T, B cells? also, it would be interesting to evaluate the immunoglobulin titer Pre and Post.
Figure 1,3, please enlarge the plots, they are small and unclear.
The male group is all teenagers, do you think their age can interfere with the findings?
Patient number 10 has, clearly, different findings than the others and her post-IgE B has not been done.
The authors have used 15-mer peptide microarray, I do have some questions, please clarify if you stored the sample pre-slit and the conditions of storage. Or if you have done the microarray test twice. Microarrays have some disadvantages such as relatively low accuracy, precision, and specificity. The sample condition interferes with the quality of the results. Have you tried to do any confirmation of the findings with any laboratory techniques?
Reviewer 4 Report
This paper describes an evaluation e present a broad analysis of IgE and IgG4 epitope binding to ten HDM allergens. The manuscript evidenced the changes in IgE and IgG4 levels before and after SLIT, however, the results aren’t conclusive because IgE levels from certain epitopes do not diminish after one-year, in addition, IgG4 levels increments only for some peptides, a major number of patients could be included in the study to evidence better the asseverations made.
Why did they not measure the levels of antibodies in shorter and longer times?
Some issues regarding the presentation of methods and results need to be clarified or mentioned in the text.
Line 95 On what basis did they determine that MFI higher than 100 was positive?
Line 116 on what basis is considered an increase or decrease in peptide reactivity if there was a difference of at least 20%?
In general, the results could be explained more simply in order to identify more quickly the importance or relevance of the results observed.
In materials and methods in section epitope microarray, more detail of microarray experiment protocol could be included.
Of the allergens that stand out in the discussion, is there anything else described in the literature, have they already been evaluated in the development of immunotherapies?
In the discussion they mention that the exclusive IgG4 epitopes could be part of a vaccine candidate but they do not discuss what has been described in the literature regarding the design and production of vaccines for the treatment of allergies.
